Corrected: Author correction

# Puma genomes from North and South America provide insights into the genomic consequences of inbreeding

Nedda F. Saremi ⓘ et al.#

Pumas are the most widely distributed felid in the Western Hemisphere. Increasingly, however, human persecution and habitat loss are isolating puma populations. To explore the genomic consequences of this isolation, we assemble a draft puma genome and a geographically broad panel of resequenced individuals. We estimate that the lineage leading to present-day North American pumas diverged from South American lineages 300–100 thousand years ago. We find signatures of close inbreeding in geographically isolated North American populations, but also that tracts of homozygosity are rarely shared among these populations, suggesting that assisted gene flow would restore local genetic diversity. The genome of a Florida panther descended from translocated Central American individuals has long tracts of homozygosity despite recent outbreeding. This suggests that while translocations may introduce diversity, sustaining diversity in small and isolated populations will require either repeated translocations or restoration of landscape connectivity. Our approach provides a framework for genome-wide analyses that can be applied to the management of similarly small and isolated populations.

---

#A full list of authors and their affiliations appears at the end of the paper.

  1

The ancestors of the puma, *Puma concolor*, also known as the mountain lion or cougar, colonized North America approximately 6 million years ago (mya)[1–3]. Although their Pliocene fossil record is sparse and felid fossil assignments have been difficult, previous mitochondrial analyses suggested that the ancestral puma lineage diverged from the extinct North American cheetah-like cat *Miracinonyx* around 3.2 mya[4]. The geographic origin of *P. concolor* remains contested, however. At sites across North America, the oldest puma fossils date to the Rancholabrean land mammal age[5], ~200 thousand years ago (kya)[6]. Analyses of mitochondrial and microsatellite data, however, estimated that the common ancestor of North American pumas lived within the last 20,000 years[7,8] and that the genetic diversity of all modern pumas traces to eastern South America[7]. The combination of genetic and fossil data were interpreted as reflecting a North American origin of the puma lineage followed by local extinction in North America during the late Pleistocene and subsequent recolonization from South America as the climate warmed after the last ice age[7,8]. Recently, however, an unequivocal puma fossil was discovered in Argentina that dates to 1.2–0.8 mya[9]. This discovery pushes back the age of the puma lineage by more than 500,000 years, and suggests that the ancestor of all living pumas may have evolved in South America rather than North America.

Today, pumas are among the most widely distributed mammals in the Western hemisphere, ranging from Canada's Yukon to the southern tip of South America (Fig. 1)[10,11]. During the 19th and 20th centuries, bounty hunting reduced, and in some cases extirpated, puma populations across North America[10], restricting them to the North American West and the southern

tip of Florida. By the middle of the 20th century, hunting quotas and some outright bans[12] allowed puma populations to increase and recolonize parts of their former range. Although some puma populations today are large and well-connected[13], others are small and fragmented (e.g., Santa Ana, CA[14]; Santa Monica Mountains, CA[15]), and/or critically endangered (e.g., Florida[16]). Many populations are experiencing increased isolation with the expansion of highways, residential developments, and agriculture[14,15].

The consequences of geographic isolation on puma genetic diversity and fitness have been well documented, particularly in Florida, where they are a federally protected subspecies commonly called the Florida panther. By the 1990s, the canonical Florida panther population in Big Cypress National Preserve was suffering from reproductive failure and phenotypic defects associated with inbreeding[16,17]. To rescue the Florida panthers from extinction, eight female pumas from Texas were released in South Florida in 1995, of which five successfully produced offspring. By 2008, the occurrence of phenotypic defects had significantly declined, survival measures had improved, and the population size increased almost threefold[16,18]. All Florida panthers genotyped since 2012 show ancestry that includes admixture with the introduced lineages[19].

Florida panthers in Everglades National Park are partially isolated from the core canonical population that persisted in Big Cypress National Preserve by a semipermeable barrier associated with hydrologic fluctuations of the Everglades. Intriguingly, during the 1990s, the Everglades panthers did not show the same high incidence of inbreeding-associated phenotypes as in the Big Cypress population. The absence of observed phenotypic defects in the Everglades population may be attributable to the release during the 1950s and 1960s of captive-bred Florida panthers with mixed Central American ancestry into Everglades National Park. The introduced individuals' ancestry was unclear at the time of release, although it was known that the captive population had greater reproductive success than wild Florida panthers[20]. The admixed ancestry of the Everglades population and potential explanation for the reproductive success of the captive population was later discovered through genotyping[21].

Genetics has a long history as a tool in wildlife conservation[22]. In traditional conservation genetics, researchers sequence a small number of genetic markers across a large sampling of the species of interest. Advances in sequencing technologies have made it possible to sequence whole genomes of non-model organisms, including species of conservation concern. While the cost of sequencing continues to decrease, sequencing whole genomes will undoubtedly remain more costly than sequencing only a handful of genetic loci. This presents a choice: whole genome data sets exchange spatial resolution for finer-scale genomic resolution, allowing researchers to test different hypotheses. Each whole genome contains a multitude of largely-independent genealogies, which provides increased power to infer past events[23,24]. In particular, the dense haplotype information provided by whole genomes is necessary to examine the very short timescales[25,26] relevant to conservation efforts.

Here, we reconstruct the last two million years of puma demographic history by generating and analyzing a draft genome from an individual sampled in the Santa Cruz Mountains (California, USA), along with nine resequenced genomes from pumas from North and South America. We confirm the recent maternal ancestry of North American pumas and describe genomic diversity in the sampled populations. We use shared tracts of homozygosity to predict the effectiveness of assisted gene flow in restoring lost genetic diversity. Finally, we analyze the genome of a Florida panther with admixed ancestry that was collected 30 years after the first release of Central American admixed pumas

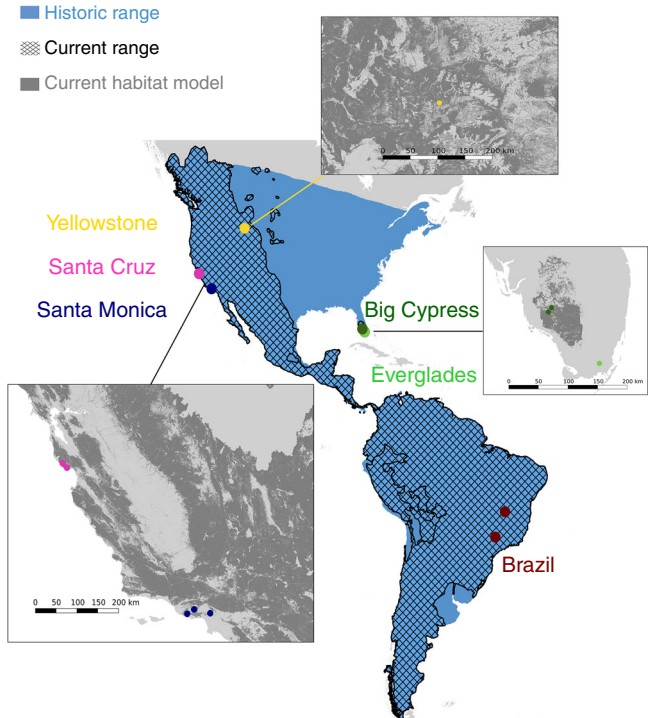

**Fig. 1** Puma range past and present. The current range of pumas (hashed) compared to their historic range (blue). Circles denote the geographic coordinates of the puma populations sampled in this study. Panels show zoom-ins of puma habitat distribution (dark gray) within the known range of the species in the contiguous United States as predicted by the USGS[59]. Current range data are from the IUCN Red List of Threatened Species[60]. Historic range data are approximated based on prior reports[12]. Base map generated with Natural Earth.

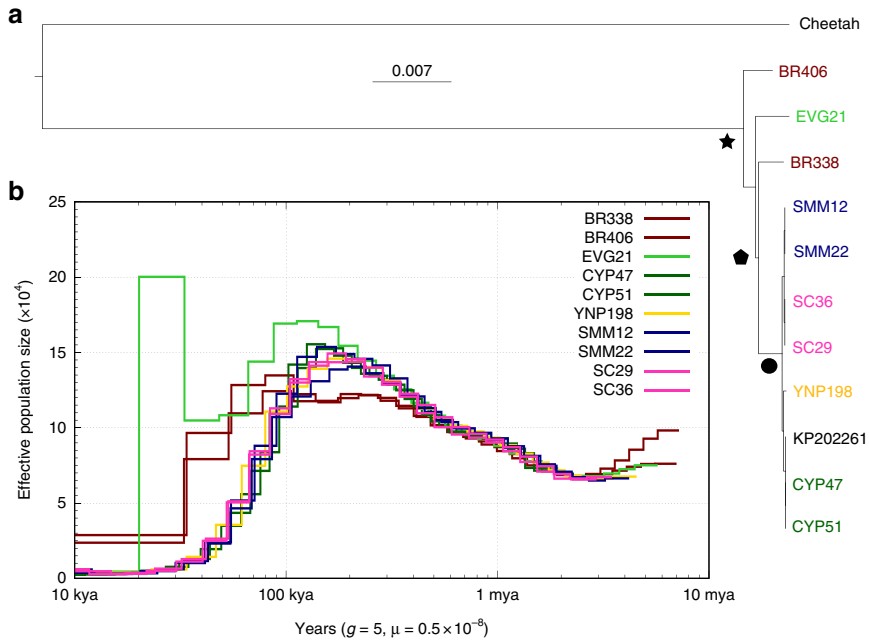

**Fig. 2** Demographic history of pumas. **a** Mitochondrial maximum likelihood phylogeny of the ten pumas in this study plus an additional puma from Big Cypress (KP202261.1) and the African cheetah (KP202271.1) as the outgroup. We calculated divergence times by determining the number of pairwise divergences between sequences and used a mitochondrial divergence rate of 1.15% bp per Myr[7,57]. We estimate a common maternal ancestor of these pumas 278,000 ± 5,639 years ago (star; 100% bootstrap support), divergence between North American and South American mitochondrial lineages 201,000 ± 1952 years ago (pentagon; 63% bootstrap support), and a common maternal ancestor of North American pumas 21,000 ± 10,412 years ago (circle; 100% bootstrap support). **b** Inferred changes in effective population size ($N_e$) over time using the pairwise sequentially Markovian coalescent (PSMC) model[29] for the ten pumas. We assume a generation time of 5 years and a per generation mutation rate of 0.5e-8 per bp per generation[61]. The PSMC model for EVG21 shows a sharp increase in inferred $N_e$ that is probably attributable to its hybrid ancestry[31].

into the Everglades. This genome allows us to assess the long-term efficacy of inter-population admixture as a means to rescue small and isolated populations from the deleterious effects of inbreeding.

## Results

**Genome assembly and variant calling**. We assembled a de novo nuclear genome for a wild male puma (SC36) from the Santa Cruz Mountains using a combination of shotgun Illumina (47× coverage), long-range linking Illumina, and Oxford Nanopore Technology (ONT) (1.2× coverage)[27] data (see Methods section). Our PumCon1.0 assembly has a BUSCO[28] score of 93.04%, a scaffold N50 of 100 Mb, and 87.6% of the genome represented on 26 autosomal scaffolds, each larger than 20 Mb (Supplementary Tables 1 and 2). Although our ONT coverage was only 1.2×, the use of these data for gap-filling recovered an additional 5.74% of the genome sequence, which we error-corrected by re-mapping the Illumina reads (Supplementary Table 1).

We obtained 27×−55× coverage whole-genome resequencing data from nine additional pumas from locations in North and South America (Fig. 1 and Supplementary Tables 3 and 4), and aligned the data to our reference assembly (PumCon1.0) for variant calling. We produced three final call sets: the first containing 8 million variable sites using the 10 pumas, the second decreased to 166,037 variable sites after filtering the first call set for linkage disequilibrium (LD), and the final call set containing 557,741 SNPs after LD filtering using the 10 pumas and the African cheetah (see Methods section).

**Demographic history**. We reconstructed puma demographic history using both mitochondrial and nuclear genomes. Analyses

of mitochondrial DNA estimate the most recent common maternal ancestor of all sampled pumas ~300 kya (Fig. 2a). North American mitochondrial haplotypes cluster together, sharing an inferred common maternal ancestor 31–11 kya. The North American mitochondrial clade excludes the Florida Everglades puma (EVG21), which has a mitochondrial ancestry that is distinct from the rest of North America, consistent with the reported mixed ancestry of this individual[16].

The nuclear genomic data revealed a similar demographic history to that inferred from the mitochondrial data, and allowed us to estimate changes in effective population size over time. Pairwise sequentially Markovian coalescent (PSMC) modeling[29] of the nuclear genomic data suggested that two puma lineages, one represented by the two Brazilian individuals and the other represented by all individuals sampled in North America, diverged by 300–100 kya (Fig. 2b and Supplementary Fig. 5), similar to the age of the oldest puma fossils in North America. Populations on both continents were largest around 130 kya, during the warmest part of Marine Isotope Stage (MIS) 5, and then declined throughout the colder MIS 4–2, with populations reaching their current small sizes by the peak of the last ice age 25–20 kya.

Our North American pumas showed a continued increase in effective population size between 500 and 200 kya, whereas the effective population size of the Brazil pumas stabilized. This increase may reflect an increase in numbers during colonization of unoccupied habitats in Central and North America, but may also be attributable to PSMC modeling overestimating effective population size when a species has divided into subpopulations[30]. To test whether this observed peak was the result of population structure, we modeled pseudo-diploid individuals using the X chromosomes of our male pumas (see Methods section). We

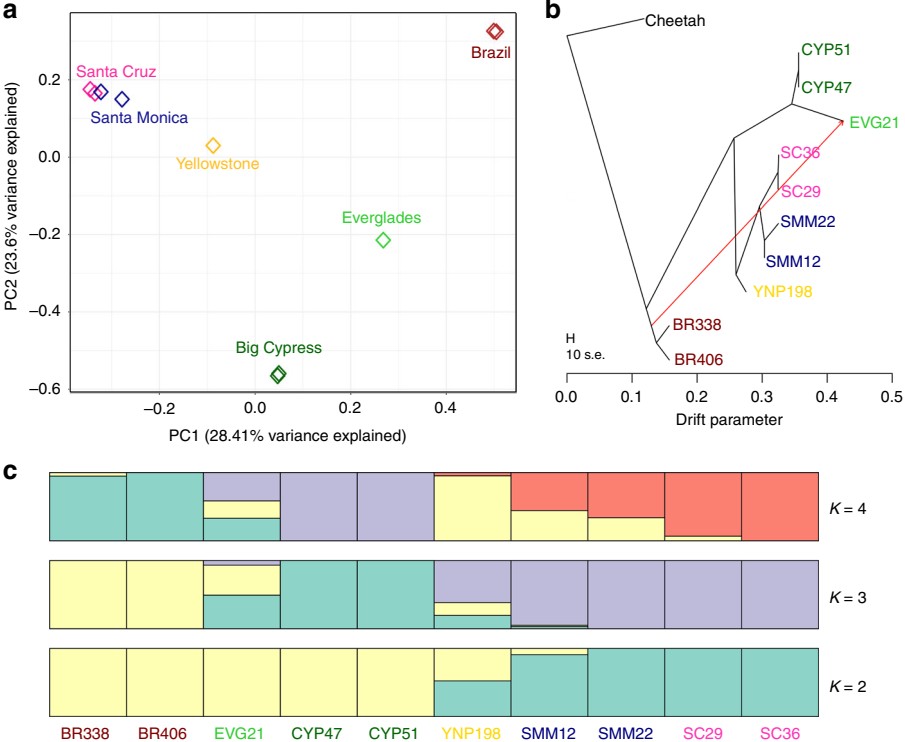

**Fig. 3** Stratification of pumas based on the geographic population. **a** Principal component analysis of 166,037 sites separates the sampled pumas based on population. The first component primarily separates South and North American pumas, while the second component distinguishes the variation within North America. All California pumas (Santa Cruz and Santa Monica) cluster closely. **b** TreeMix[62] analysis, using the African cheetah as the outgroup, indicates the best tree separates pumas based on population and includes one migration event (weight = 0.453911) from the branch of South American diversity into the admixed Everglades puma (EVG21). **c** The mean of ten permuted matrices of STRUCTURE[63] analysis for each of K = 2 through 4, performed using CLUMPP[64]. Both delta K and L(K) values indicated that K = 3 was the best K (Supplementary Fig. 10)[65].

found evidence of a cessation of gene flow between all Brazil–North America pseudo-diploid male puma pairs by at least 100 kya, as shown by the sharp increase in the inferred effective population size ($N_e$), signifying no coalescent events occurred more recently than the estimated divergence time (Supplementary Fig. 7). The divergence dates obtained from the pseudo-diploid X chromosome PSMC analysis overlapped the time at which North and South American inferred $N_e$ began to differ in the autosomal PSMC model. Thus, structure alone was not the reason behind the observed increase in $N_e$ during this time. The spike in effective population size observed for EVG21 probably does not reflect the coalescent process within a single population, but is instead consistent with mixed ancestry comprised of two divergent lineages[31].

**Population structure**. We used the nuclear genomic data to characterize genetic structure among puma populations (Fig. 3). We performed principal component analysis (PCA) of 166,037 LD-filtered SNPs and found evidence of a geographic pattern (Fig. 3a). The first two axes of the PCA, which explain 52% of the genetic variance, separated North and South America and revealed a gradient of relatedness from east to west across North America. The Everglades puma (EVG21) fell between the Big Cypress and Brazil populations, consistent with this individual's known history of admixture. Pumas sampled from the same population clustered together.

We estimated a consensus nuclear phylogeny from 557,741 SNPs from the LD-filtered data set that included ten pumas and the African cheetah. This analysis found further evidence of structure, with the highest likelihood tree including a single

migration event from a South (or Central) American lineage into the Everglades lineage (Fig. 3b and Supplementary Figs. 8 and 9). Finally, our cluster assignment tests based on the puma only LD-filtered SNPs also partitioned the data geographically, first separating out the two California populations at K = 2, and then the Florida and Brazil populations at K = 3 (Fig. 3c and Supplementary Fig. 10). Notably, EVG21 shares ancestry with both Florida and Brazil at all K values (Supplementary Fig. 11).

We note that the discrete populations identified in these analyses could simply reflect the spatial sampling of our data set. Spatially structured sampling can cause analyses to report distinct populations even when no discrete population structure exists[32]. This artifact is particularly likely to occur when geographically widespread samples are taken from well connected species where limited dispersal results in the accumulation of local genetic variants, resulting in genetic isolation by distance[33]. However, the observed geographic structure could also be the result of discrete genetic structure due to population isolation. Some puma populations have experienced persecution and degradation of their habitat, resulting in limited gene flow between populations[16,34–36]. These isolated populations would show increased divergence over time, resulting in geographic structure.

**Heterozygosity and inbreeding**. To examine the extent of inbreeding in our puma samples, we estimated for each individual average genome-wide heterozygosity and identified runs of homozygosity (ROH) across the 26 largest autosomal scaffolds (Fig. 4, see Methods section). We focused our analyses on ROH > 2 Mb, as we were able to call these longer tracts with high confidence. Although genetic drift is a dominant evolutionary force

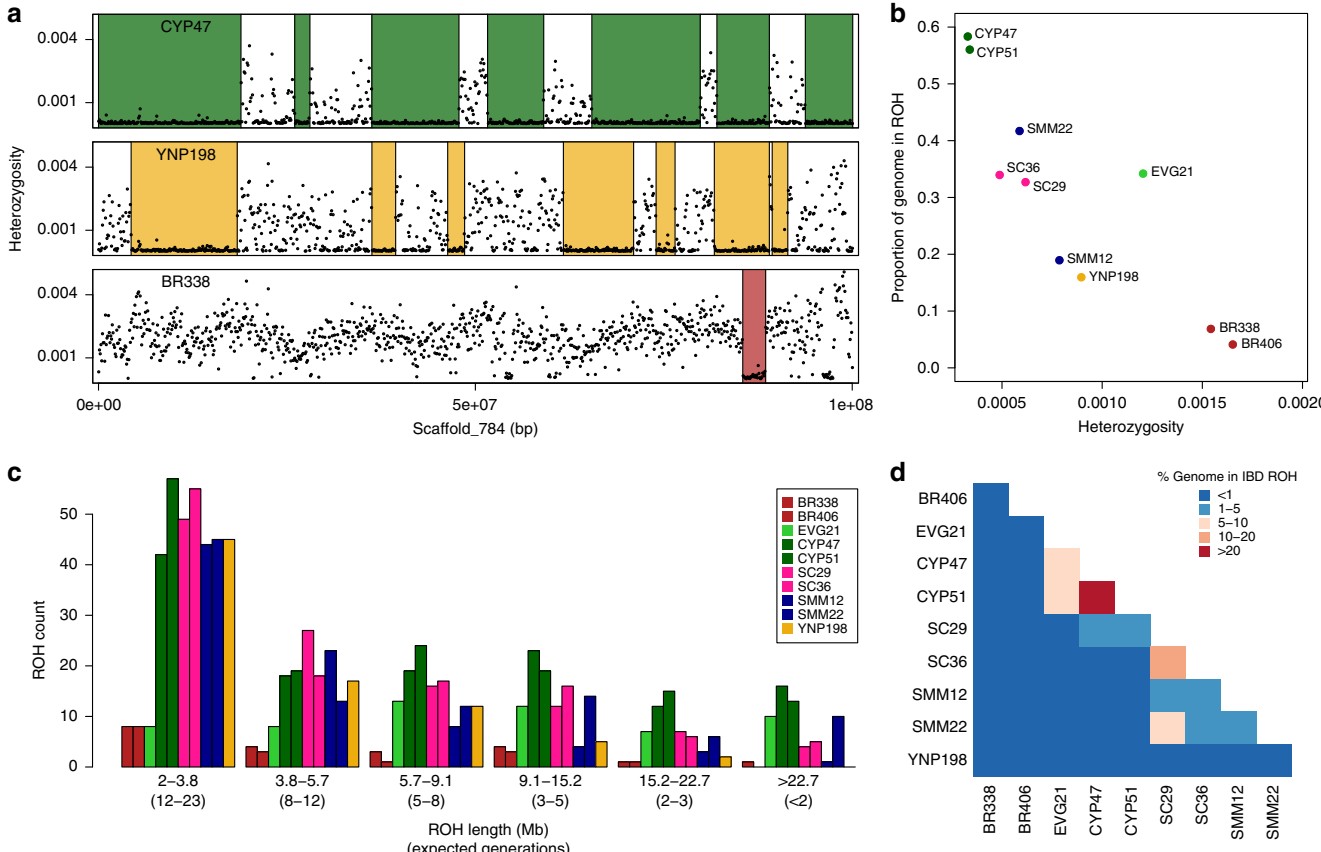

**Fig. 4** Heterozygosity and runs of homozygosity. **a** Sliding window heterozygosity (black dots) and called ROH (colored boxes) across a single scaffold for three pumas from three different populations (Big Cypress, Yellowstone, and Brazil). Plots for all pumas are provided as Supplementary Fig. 12. **b** Average genome-wide heterozygosity versus the proportion of the genome in ROH for the ten pumas sequenced. **c** Distribution of lengths of ROH. The length in Mb is indicated, as is the associated expected number of generations since the individual's maternal and paternal lineages shared a common ancestor. **d** Heat map showing the percent of the genomes that are in ROH that are shared IBD between pairs of pumas (Supplementary Table 6).

in small populations, the strong correlation among linked sites that is characteristic of ROH > 2 Mb requires close inbreeding, and would not be observed due to genetic drift alone[37,38]. The distribution of ROH across the genome varied among scaffolds and individuals (Fig. 4a and Supplementary Fig. 12), as did average genome-wide heterozygosity and proportion of the genome in ROH (Fig. 4b). The two pumas from Brazil were the least inbred, with the highest heterozygosity and smallest proportions of their genomes in ROH. Conversely, the Big Cypress panthers sampled prior to the 1995 genetic rescue were the most inbred, with the lowest heterozygosity and the largest proportions of their genomes in ROH, consistent with the phenotypic defects recorded in these individuals[16]. The other North American pumas fell between these two extremes. Of the two individuals from the Santa Monica Mountains, SMM12 appeared to be less inbred than SMM22, with higher heterozygosity and a lower proportion of its genome in ROH. This is consistent with their origins, as genetic analysis suggests that SMM22 was likely born in the small and more isolated Santa Monica Mountains population south of US 101 freeway, whereas SMM12 was first observed in the larger and more connected population north of US 101 and dispersed into the Santa Monica Mountains as a subadult[15].

EVG21, the admixed Florida panther from the Everglades population, was an outlier in the general correlation between heterozygosity and proportion of the genome in ROH. The proportion of EVG21's genome in ROH was high relative to the expectation based on its average genome-wide heterozygosity. This is consistent with both ancestral admixture resulting in a

more diverse genetic background and close inbreeding leading to long tracts of homozygosity (Supplementary Fig. 13).

To better explore inbreeding history, we examined the distribution of ROH tract lengths in each puma. We correlated those lengths with the expected number of generations since the individual's maternal and paternal lineages shared a common ancestor using an estimated average recombination rate from the domestic cat of 1.1 cM per Mb[39] and the equation $g = 100/(2rL)$, where $g$ is the time in generations, $r$ is the recombination rate, and $L$ is the length of the ROH tract in Mb[26,40] (Fig. 4c). Long ROH (>15.2 Mb) occur due to close inbreeding (a common ancestor <3 generations ago). Short ROH (<5.7 Mb) occur due to shared ancestors further back in time (>8 generations ago). All North American pumas sampled had a large number of short ROH, indicating that these populations were small in the recent past (8–23 generations ago). The puma from Yellowstone had mostly short ROH and a small number of intermediate and long ROH, consistent with a population that was small in the recent past, but that does not suffer from a considerable amount of close inbreeding in recent generations. The pumas from the Santa Cruz and Santa Monica Mountains had patterns similar to the Yellowstone puma, except they had additional long ROH, suggesting that these populations are experiencing close inbreeding. The Big Cypress panthers each had many long ROH, which we estimated to reflect shared ancestors within the last three generations.

The admixed Everglades panther, EVG21, had a small number of short ROH, similar to the Brazilian pumas, but had mostly

long ROH, similar to the more inbred Florida individuals. This combination can be attributed to EVG21's complex history of admixture and inbreeding. EVG21 has historic admixture, and is the offspring of an inbreeding event—the sire of EVG21 was also EVG21's half brother[16] (Supplementary Fig. 4). The peak of the ROH length distribution for EVG21 occurs at 5.7–9.1 Mb, indicating that EVG21's maternal and paternal lineages shared a common ancestor as far back as 5–8 generations, shortly after the admixture event that occurred 6–9 generations prior[16].

Although the sampled North American pumas all have long ROH, these tracts were generally not identical by descent (IBD) between individuals (Fig. 4d). Long ROH that are also shared IBD between individuals are concerning because they represent regions of the genome with no genetic diversity in the four haplotypes analyzed. Of the pumas sequenced, only the two individuals from Big Cypress (CYP47 and CYP51) shared a considerable proportion (36%) of their genomes in ROH that are IBD between two individuals. The pumas from the Santa Cruz Mountains (SC29 and SC36) shared 12% of their genomes in IBD ROH, whereas the pumas that originate from different areas in and near the Santa Monica Mountains (SMM12 and SMM22) shared only 4%. Individuals from the Santa Cruz and Santa Monica Mountains shared between 3% and 5%. While most sampled North American populations show signs of close inbreeding, different populations are fixed for different variants and considerable genetic variation still exists when considering the species as a whole.

## Discussion

We present a draft assembly of a puma genome, which we use to reconstruct the demographic history of the species and measure genome-wide heterozygosity and ROH, the latter of which is less practical with lower-quality or reference-guided genome assemblies. Our assembly strategy combined short-read Illumina data with long-read data from ONT to generate a scaffold N50 of 100 Mb, making this one of the most contiguous wild felid genomes assembled to date.

Our analyses of ten complete puma genomes revealed the dynamic history of a once widespread species whose population size is now reduced across much of its range. We showed that extant North American pumas are descended from a population that dispersed northward from South America by at least 200 kya, consistent with the age of the oldest puma fossils in North America. Previously, the incomplete fossil record paired with divergence estimates based on rapidly evolving microsatellites and partial mitochondrial genomes led to the hypothesis of a North American origin of the species, followed by a late Pleistocene local extinction in North America and then a recolonization from South America within the last 20,000 years[7,8]. Our results using complete nuclear and mitochondrial genomes are consistent with previous genetic analyses in that we show that North American pumas represent a subset of puma genetic diversity. However, the nuclear genomic data suggest that the lineage leading to North American pumas diverged from South American pumas ~300–100 kya, considerably older than the 20 kya inferred previously. While we are unable to exclude the possibility of a local late Pleistocene extinction in North America followed by a recolonization from an unsampled lineage elsewhere in South or Central America, we argue that this nuclear genomic data in combination with a recently identified puma fossil in South America that dates to 1.2–0.8 mya[9] supports a simpler demographic hypothesis in which the puma lineage originates in South America, disperses into North America by 300–100 kya and persists there to the present day. We note that new fossils or genomic data from late Pleistocene aged pumas or

pumas from other locations in South and Central America will be necessary to test this demographic hypothesis.

If true, the new model for puma demographic history means that pumas would have been present in North America for at least one complete glacial/interglacial cycle, indicating that pumas were capable of surviving in a broad range of habitats and environments. This hypothesis is supported by data from living pumas, which, despite a preference in North America for mountainous habitats, are also known to occupy grassland habitats in South America, such as Patagonia[10]. Differences in habitat selection between the two continents probably reflect a long history of competition with a diverse carnivore guild on both continents. For example, jaguars are better adapted than pumas to living in habitats that flood periodically[41], and predation by wolves in North America probably precludes pumas living in open habitat without escape terrain[42].

Intriguingly, North American pumas share a common maternal ancestor around the peak cold period of the last ice age, ~20 kya. This period is associated with a reduction of available habitat across the continental United States, as the coalesced Laurentide and Cordilleran glaciers covered much of present-day Canada and the Upper Midwestern United States[43]. Forests would have been reduced significantly at that time, as would available habitat for the smaller prey preferred by pumas, providing a potential mechanism for a reduction in puma population size around that time.

The recent history of pumas is marked by human persecution and encroachment on their habitat, resulting in small and isolated populations that are susceptible to loss of genetic diversity and predisposed to inbreeding. Over many generations, without the input of novel variation from migrants, isolated populations can accumulate local genetic variation while losing overall genetic diversity. Loss of genetic diversity may be a common situation for top predators, as their population densities are usually low and successful migrants are infrequent. Consequently, even moderate levels of fragmentation will affect their genomic diversity. While pumas in South America currently experience less habitat degradation than pumas in North America, pumas in South America will likely face further habitat loss and fragmentation as rapid human population growth and land development continues on the continent[13]. The result may be small, isolated populations in South America similar to those currently seen in North America. Thus conservation efforts and findings taken from isolated populations in North America may need be applied in the future to other parts of the puma range.

In North America, pumas were hunted extensively, resulting in low population densities in many areas of their range[10]. Hunting was so severe until regulations were put in place during the mid 20th century that pumas likely experienced a population bottleneck. All North American pumas sampled in this study exhibit short ROH that date to approximately the early 20th century, indicative of small effective population sizes during the time when hunting was severe.

In many areas of North America, including California and Florida, large-scale hunting was followed by shrinking habitat availability, resulting in small, isolated populations[15,44,45]. Our sampling focused on populations in North America that are known to be isolated and, as such, our results highlight the genomic consequences of this isolation—reduced diversity and signatures of close inbreeding. Pumas in the isolated populations of Big Cypress (CYP), Santa Monica Mountains (SMM), and Santa Cruz Mountains (SC) all have many ROH of all length categories, indicating ongoing inbreeding as a result of continued small population sizes. In contrast, the Yellowstone individual had a similar number of short ROH to these more isolated populations, but fewer long ROH. This pattern is consistent with

the known history of hunting and habitat availability in the Yellowstone area. Pumas in the Yellowstone area were hunted to low densities into the mid 20th century[10], but today Yellowstone National Park is a large protected area surrounded by wildlands. This connectivity between the Park and wildlands facilitated the recovery and maintenance of genetic diversity in the local puma population once hunting pressures were reduced.

Florida panthers are among the most well-studied populations of pumas, especially with regard to the phenotypic manifestations of isolation and inbreeding. The 1995 introduction of pumas from Texas, the most geographically proximate population to the Florida panthers, is widely regarded as a successful genetic rescue via translocation. However, Florida panther genetic diversity in the Everglades population had been bolstered several decades earlier, when seven individuals were released into Everglades National Park from a captive facility where pumas from Central America had been included in the breeding population[20]. One Florida panther that we sequenced, EVG21, is admixed, having both Floridian and Central American ancestry. Her genome is a combination of regions with comparatively high heterozygosity, similar to that observed in the Brazilian pumas, and long ROH, similar to the highly inbred Florida panthers. The distribution of the lengths of ROH suggest that her maternal and paternal lineages shared a common ancestor that lived shortly after the release of the admixed pumas into the Everglades population. This suggests that the genomic consequences of inbreeding happen quickly, with much of the gains from the genetic rescue being quickly erased. EVG21's genome provides evidence that when the population is small, it is likely that an individual's parental lineages will share a very recent common ancestor, even after genetic rescue through admixture (Supplementary Fig. 13). Thus, a consistent effort is required to maintain the benefits of translocation.

In many areas of the current puma range, human land use has reduced the connectivity that is critical to recovery and maintenance of healthy populations. Despite these barriers, gene flow among neighboring populations can be facilitated by enhancing landscape connectivity through coordinated land use planning and by adding bridges or underpasses across freeways[46]. Although pumas are capable of traveling long distances, large roads are a major barrier to their movement[14,47]. A model of population dynamics in the Santa Monica Mountains that incorporated landscape connectivity and its effects on genetic diversity predicted a high probability of extinction (99.7%) within 50 years after survival rates first began to decrease due to inbreeding, unless connectivity was increased[48]. Our genomic analyses of the samples from the Santa Monica Mountains also support the effectiveness of population connectivity. The two pumas sequenced from the region (SMM12 and SMM22) both currently reside in the small subpopulation south of US 101 freeway. However SMM12 migrated into the subpopulation from north of US 101[15], a larger area that shows greater connectivity to surrounding regions. Migrations between these two areas are now rare, but the two subpopulations were probably part of a larger panmictic population prior to the existence of US 101. The genomic analysis of ROH highlighted that SMM22 had an increased number of large ROH relative to SMM12, consistent with SMM22 originating in a population that is smaller and more isolated. The examination of IBD ROH between SMM12 and SMM22 showed that only 4% of their genomes are in ROH that are IBD between the two individuals. In contrast, individuals that originated from the same population have a much larger proportion of their genomes in IBD ROH (e.g., 12% for SC29 and SC36). This indicates that while inbreeding has reduced diversity in a considerable proportion of the genomes of individuals within small populations, these low diversity regions are generally not shared between populations. Thus, reconnecting the populations on either side of US 101, as currently proposed via a wildlife crossing over the freeway, would help restore the lost genetic diversity.

Genome-scale data sets have the potential to inform conservation planning. Our results highlight how whole genome data can provide new insights when compared to traditional conservation genetic techniques. For instance, measures of average heterozygosity are the most commonly used metrics to characterize the genetic health of a species, as estimates are relatively simple to generate and are easily comparable among organisms. However, average heterozygosity provides only a narrow insight into the health and genetic potential of a species[49]. While in some species average genome-wide heterozygosity is highly correlated with the level of inbreeding estimated using ROH[26], in systems with admixture, average heterozygosity estimates can be deceptive, as demonstrated with our admixed Everglades puma (EVG21). The heterozygosity of EVG21 is almost as high as the Brazilian pumas, but EVG21 has a large portion of her genome in ROH. We would infer two very different genetic conditions when considering each metric separately, and thus both heterozygosity and proportion of the genome in ROH should be considered in assessing genomic health. Finally, knowledge of shared ROH, an analysis which can only be done with very high density markers across the genome, is critical when designing mitigation plans, as this analysis predicts whether enhancing connectivity would restore lost genetic diversity and helps identify potential candidates for translocation. In this context, this study can serve as a template for future conservation genomic research targeting species living in small, isolated populations.

## Methods

**Assembly and annotation of the puma reference genome**. We captured and drew blood from a wild, male puma (SC36) who lived in the Santa Cruz Mountains in California, USA in accordance with guidelines and regulations of local governing bodies (Supplemental Methods). We extracted DNA and generated a combination of short-read paired-end, proximity-ligation, and long-read data (Supplemental Methods). We assembled a de novo shotgun assembly using trimmed paired-end short reads, and scaffolded the assembly using proximity-ligation data[50] (Supplementary Fig. 1). We performed gap-filling on the scaffolded genome assembly using long-read data and corrected the newly gap-filled sequence using the short-read paired-end libraries (Supplementary Methods and Supplementary Table 1). Given that the puma used for the shotgun assembly was a male, we identified three X chromosome scaffolds in a female genome assembly (SMM13) and added these scaffolds (scaffolds X1, X2, and X3) to the assembly for SC36 (Supplementary Fig. 2 and Supplementary Methods).

We assessed this final version of the genome (PumCon1.0) by alignment to the domestic cat genome (GCA_000181335.4) (Supplementary Fig. 3). We used the genome assessment tool BUSCO[28] (version 2.0.1) to evaluate genome completeness based on a set of conserved single-copy orthologous genes (human gene set; $n = 4104$). In the PumCon1.0 genome, 93.0% of these genes are complete and present in a single copy only (Supplementary Table 2). The final genome assembly is 2,432,985,507 bp in length with an N50 of 100.53 Mb, 178,994 gaps, and 114,069,924 Ns. We focused further analyses on the 87.6% of the genome that is represented on 26 autosomal scaffolds, each larger than 20 Mb.

We generated and sequenced a cDNA library from whole blood collected from a wild puma (SC85) from the Santa Cruz Mountains (Supplementary Methods). The PumCon1.0 genome was annotated by NCBI according to the NCBI Eukaryotic Genome Annotation Pipeline[51] using our cDNA data and a publicly available data set generated from a wild puma from Arizona (SAMN02885420, SRX633288).

**Additional puma genomes**. We generated genomic data for a total of 11 pumas (Supplementary Tables 3 and 4). We used data from one female from the Santa Monica Mountains to assemble the X chromosome (SMM13). The other ten pumas, including the individual used for the genome assembly (SC36), were used in a panel for analysis of demographic history, population structure, and inbreeding. The ten pumas that formed our panel were: two pumas from the Santa Cruz Mountains in Northern California (SC29, SC36), one puma from Yellowstone National Park (YNP198), two pumas from the Big Cypress National Preserve that were part of the canonical (pre-Texas admixture) Florida panther population (CYP47, CYP51), one puma from the population that lived in Everglades National Park in Florida that was the admixed descendent of a canonical Florida panther

and a puma of Central American ancestry that was released into the Everglades decades prior to the Texas panther introduction[16] (EVG21), two pumas from the Santa Monica Mountains in Southern California (SMM12, SMM22), and two pumas from eastern Brazil (BR406, BR338). Capture, handling, and sampling of all pumas involved in this study were approved by the appropriate governing bodies (Supplementary Methods). We generated ~30× coverage of short-read data for the 11 pumas described above (Supplementary Methods), and downloaded shotgun sequencing data for the African cheetah[52] (SRR2737512-SRR2737518) to use as the outgroup for our analyses.

To perform variant calling and filtering for the puma genomes, we mapped adapter-trimmed resequencing data and cheetah SRA data to the PumCon1.0 genome, including the mitochondrial scaffold (Supplementary Methods). Due to the high number of nuclear mitochondrial DNA segments (NUMTs) in felids[53], we sought to decrease mismappings of authentic mitochondrial DNA in our data to NUMTs. We generated three sets of genotypes: two sets comprised the ten pumas (one set was LD filtered and the other was not LD filtered), and a third included the ten pumas plus the cheetah (LD filtered). For all variant files, we masked and removed sites that were not biallelic SNPs, and did not pass our filtering criteria (Supplementary Methods). We removed mitochondrial and X chromosome related scaffolds, and used only autosomal scaffolds for further analyses (scaffold Mt, X1, X2, X3, 869, 1862) (Supplementary Methods).

The non-LD filtered puma-only variant file contained 8,212,535 SNPs. The final LD-filtered puma-only variant file contained 166,037 SNPs. The LD-filtered puma and cheetah variant file contained 557,741 SNPs. The larger number of variants in the puma and cheetah file is due to sites where the cheetah carries two of the alternate allele while all pumas carry the reference allele. Using the non-LD filtered SNP calls from the puma-only data set, we generated a fasta file for each sample, masking both failed SNP sites and failed individual genotypes to Ns (Supplementary Methods).

**Mitochondrial genome assemblies and phylogeny inference**. We assembled an initial mitochondrial sequence for SC36 using short and long read data that mapped to the available puma reference mitochondrial sequence (KP202261.1) (Supplementary Methods). We then used adapter-trimmed Illumina shotgun data to assemble the mitochondrial genome sequences of the remaining nine pumas. We used the iterative assembler mia[54], with the SC36 mitochondrial sequence as the reference. The coverages of these mitochondrial assemblies ranged from 35× to 138×. We annotated the mitochondrial genomes using MITOS[55].

We constructed a maximum likelihood phylogeny using a single partition data set, and a GTR + GAMMA substitution model using the program RAxML[56] (Supplementary Methods), including our ten assembled puma mitochondrial genomes, the available puma reference mitochondrial sequence (KP202261.1), and a cheetah mitochondrial sequence (KP202271.1) as the outgroup. We estimated divergence times using a prior composite estimate of the feline mitochondrial divergence rate of 1.15% bp per million years[7,57] (Supplementary Methods).

**Demographic history**. We used the pairwise sequentially Markovian coalescent (PSMC) model[29] to estimate the historical effective population sizes of puma populations (Supplementary Methods). We performed one hundred replicate bootstraps for each individual per the software instructions (Supplementary Fig. 5). We also ran the PSMC tool on outbred regions of the genome, identified by being void of ROH, and saw no considerable difference from the full genome results (Supplementary Methods and Supplementary Fig. 6). In addition, we investigated the divergence time between our North and South American male pumas by running PSMC modeling of X chromosome pseudo-diploid sequences of each male North American puma with that of either of the two male Brazilian pumas (Supplementary Fig. 7).

**Population structure**. We ran principal component analysis on the LD-filtered variant file for the ten pumas, which consisted of 166,037 SNPs (Supplementary Methods). We constructed a tree to show population splits, both with and without the admixed sample EVG21, using the 557,741 SNPs in the LD-filtered variant file that included the cheetah outgroup (Supplementary Methods). We inferred population structure of the pumas using the LD filtered variant file with 10 pumas and 166,037 SNPs (Supplementary Methods and Supplementary Figs. 10 and 11).

**Genome-wide heterozygosity and runs of homozygosity**. We calculated genome-wide heterozygosity using three different methods: two reference-based and one non reference-based (Supplementary Methods and Supplementary Table 4).

We used a hidden Markov model (HMM) to identify ROH by identifying transitions between inbred and outbred regions of the genome (https://github.com/russccd/Heterozygosity_HMM). We estimated HMM model parameters from the data and used the filtered fasta files with IUPAC codes as input (Supplementary Methods and Supplementary Table 5). We converted the ROH tract lengths to generations using an estimated average recombination rate from the domestic cat (Supplementary Methods).

We used the sliding window approach in PLINK[58] (version 1.90b4.4) to identify ROH for comparison with our ROH HMM. Even with relaxed parameters, PLINK still tended to break up long tracts (Supplementary Fig. 14). Since accurate estimates of tract lengths were key to our inbreeding analysis, we used the ROH called by our ROH HMM program for further analyses.

We observed a low frequency of short ROH in the genome of the admixed Everglades panther (EVG21) relative to the other Florida panthers. We hypothesized that, because an individual cannot have a shared maternal and paternal ancestor that dates to before the admixture event, admixture in previous generations may have prevented the formation of short ROH. To test our hypothesis, we used an HMM to classify tracts of ancestry in the IUPAC coded fasta file of EVG21 into three types: pure Central/South American ancestry, pure Floridian ancestry, and mixed Central/South American and Floridian ancestry. The genome of EVG21 was composed of 21.98% Central/South American ancestry, 28.24% Floridian ancestry, and 49.58% mixed ancestry based on the HMM (Supplementary Fig. 15). Using ROH greater than 2 Mb that we identified with the ROH HMM, we classified each ROH as one of the three ancestry types. The results of this analysis classified all ROH as either pure Florida or pure Central/South American ancestry. We saw no ROH that were classified as being of mixed ancestry. Thus, admixture effectively prevents the formation of mixed ancestry ROH (Supplementary Figs. 15 and 16 and Supplementary Methods).

We estimated the proportion of the ROH that are shared between pairs of pumas by finding genomic regions where ROH overlap between pairs of samples (Supplementary Methods). For each pair of pumas we calculated the proportion of the genome that occurs in ROH that are IBD (Supplementary Table 6).

**Reporting summary**. Further information on research design is available in the Nature Research Reporting Summary linked to this article.

## Data availability

The datasets generated for this study are available in public repositories. Sequence data used for the genome assembly have been deposited in the SRA with the accession numbers SRR7148342-SRR7148354 [https://www.ncbi.nlm.nih.gov/Traces/study/?acc=SAMN08662999]. The PumCon1.0 genome is available on GenBank with the accession number GCF_003327715.1. The RNA-Seq data is available on the SRA with the accession number SRX4067841. Sequencing reads for the panel of pumas have been deposited in the SRA with the accession numbers SRR7639695, SRR7639696, SRR7542886-SRR754288, SRR7660678-SRR7660679, SRR7664677-SRR7664678, SRR7956993-SRR7956994, SRR7610940-SRR7610941, SRR7661934-SRR7661935, SRR7690239-SRR7690240, SRR7543017-SRR7543018, SRR7537344-SRR7537345, and SRR7148342-SRR7148354. Annotated mitochondrial assemblies for the ten pumas are available on GenBank with the accession numbers MH807447, MH814703, MH814704, MH814705, MH814706, MH814707, MH818219, MH818220, MH818221, and MH818222. All other relevant data is available upon request.

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

## Acknowledgements

We thank Paul Houghtaling for helping to collect samples, R. Miotto, E. Amorim, J. May, CENAP/ICMBio/Brazil and AMC/Brazil for access to samples, and S. Webber and C. Scelfo-Dalbey for assistance in generating sequencing data. The authors would like to acknowledge support from Science for Life Laboratory, the National Genomics Infrastructure, and UPPMAX for providing assistance in massive parallel sequencing and computational infrastructure. Sequencing was also performed by the Laboratório de Biotecnologia Animal at the Universidade de São Paulo in Brazil, UC San Diego Institute for Genomic Medicine Genomics Center, UC Berkeley Vincent J. Coates Genomics Sequencing Laboratory, and UC Santa Cruz Ancient and Degraded Processing Center. Funding was provided by the Blue Foundation, and by a grant to C.C.W. from the Gordon and Betty Moore Foundation. C.C.W. was funded in part by NSF grants 1255913 and 0963022. B.S., M.A.S., N.F.S., and R.K.W. were funded by a grant from the University of California Office of the President. N.F.S. was funded in part by T32 HG008345/HG/NHGRI NIH HHS/United States. L.D. was funded by Formas grant 2015-676. D.R.S. was funded in part by Yellowstone Forever. R.B.C.-D. was funded by NIH-R35GM128932. B.S. and R.E.G. were funded in part by NSF DEB-1754551. E.E., L.L.C., and H.V.F. were supported by funds from CNPq/Brazil and INCT-EECBio/Brazil. Portions of this manuscript were prepared while W.E.J held a National Research Council Research Associateship Award at the Walter Reed Army Institute of Research and the published material reflects the views of the authors and should not be construed to represent those of the Department of the Army or the Department of Defense.

## Author contributions

B.S., C.C.W., and R.E.G. conceived and designed the study. C.C.W., C.V., D.P.O., D.R.S., E.E., H.V.F., J.A.S., L.D., L.L.C., P.M.S.V., R.K.W., S.J.O., S.P.D.R., and W.J. provided samples or data for this work. A.B., B.O., H.J.M. and P.M.S.V. performed laboratory work. B.S., R.B. C.-D., and R.E.G. supervised the analysis. J.A.C., M.A.S., N.F.S., and R.B.C-D analyzed the data. B.S., C.C.W., E.E., M.A.S., N.F.S., R.B.C.-D., R.E.G., R.K.W., S.J.O., and W.J. interpreted the results. B.S., M.A.S., and N.F.S. wrote the paper. All authors edited the paper.

## Competing interests

The authors declare no competing interests.

## Additional information

Nedda F. Saremi [1,19], Megan A. Supple [2,19], Ashley Byrne [3], James A. Cahill [2,16], Luiz Lehmann Coutinho [4], Love Dalén [5], Henrique V. Figueiró [6], Warren E. Johnson [7,17], Heather J. Milne [2], Stephen J. O'Brien [8], Brendan O'Connell [1,18], David P. Onorato [9], Seth P.D. Riley [10,11], Jeff A. Sikich [10], Daniel R. Stahler [12], Priscilla Marqui Schmidt Villela [13], Christopher Vollmers [1], Robert K. Wayne [11], Eduardo Eizirik [6], Russell B. Corbett-Detig [1], Richard E. Green [1], Christopher C. Wilmers [14] & Beth Shapiro [2,15]*

[1]Department of Biomolecular Engineering, University of California, Santa Cruz, 1156 High Street, Santa Cruz, CA 95064, USA. [2]Department of Ecology and Evolutionary Biology, University of California, Santa Cruz, 1156 High Street, Santa Cruz, CA 95064, USA. [3]Department of Molecular, Cell, and Developmental Biology, University of California, Santa Cruz, 1156 High Street, Santa Cruz, CA 95064, USA. [4]Laboratório de Biotecnologia Animal, Departamento de Zootecnia, ESALQ, Universidade de São Paulo, Caixa Postal 09, Piracicaba, SP 13418-900, Brazil. [5]Department of Bioinformatics and Genetics, Swedish Museum of Natural History, P.O. Box 50007, Stockholm 10405, Sweden. [6]Escola de Ciências, Pontifical Catholic University of Rio Grande do Sul, Avenida Ipiranga, 6681-Partenon, Porto Alegre-RS 90619-900, Brazil. [7]Smithsonian Conservation Biology Institute, Smithsonian Institution, 600 Maryland Avenue SW, Washington, DC 20002, USA. [8]Theodosius Dobzhansky Center for Genome Bioinformatics, Saint Petersburg State University, 41 Sredniy Prospekt, Saint Petersburg 199004, Russia. [9]Fish and Wildlife Research Institute, Florida Fish and Wildlife Conservation Commission, 298 Sabal Palm Road, Naples, FL 34114, USA. [10]Santa Monica Mountains National Recreation Area, 401 West Hillcrest Drive, Thousand Oaks, CA 91360, USA. [11]Department of Ecology and Evolutionary Biology, University of California, Los Angeles, 610 Charles E. Young Drive South, Los Angeles, CA 90095-1601, USA. [12]Yellowstone Center for Resources, P.O. Box 168, Yellowstone National Park, WY 82190, USA. [13]EcoMol Consultoria e Projetos, Avenida Limeira, 1131- Areiao, Piracicaba-SP, Brazil. [14]Environmental Studies Department, University of California, Santa Cruz, 1156 High Street, Santa Cruz, CA 95064, USA. [15]Howard Hughes Medical Institute, 400 Jones Bridge Road, Chevy Chase, MD 20815, USA. [16]Present address: Laboratory of Neurogenetics of Language, Rockefeller University, 1230 York Avenue, New York, NY 10065, USA. [17]Present address: Walter Reed Biosystematics Unit, Smithsonian Institution, 4210 Silver Hill Road, Suitland, MD 20746, USA. [18]Present address: Department of Medical Informatics and Clinical Epidemiology, Oregon Health and Science University, 3181 S. W. Sam Jackson Park Road, Portland, OR 97239-3098, USA. [19]These authors contributed equally: Nedda F. Saremi, Megan A. Supple. *email: beth.shapiro@gmail.com

