## [Peer Review File · Nature Communications]

Editorial Note: This manuscript has been previously reviewed at another journal that is not operating a transparent peer review scheme. This document only contains reviewer comments and rebuttal letters for versions considered at Nature Communications .

Reviewers' comments:

Reviewer #1 (Remarks to the Author):

In this manuscript, Saremi, Supple, et al. developed a chromosome-level genome assembly and resequenced the genomes of ten pumas across the Americas. This work provides the potential for important preliminary information on puma evolutionary history by conducting full genome sequencing on ten pumas across the range. Importantly, this marks the first published genome of *Puma concolor* which in itself is an important contribution to the scientific literature. This manuscript represents a monumental effort and has implications for the ecology, evolution, and conservation of pumas. A major strength of this manuscript is the vast quantity of sequence data. The construction of the genome was comprehensive, accounting for sex chromosomes, combining long reads with shorter high coverage Illumina runs.

In addition, the genomes of the Florida panther provide a unique opportunity to evaluate the effects of a genetic rescue event on the genome level. As noted previously, the genomes in themselves are a valuable addition to the literature. In addition, the authors have done a great job with the revision providing not only a clearer explanation of how results can be interpreted but highlighting alternative explanations. The response letter was well done and thoroughly addressed reviewer concerns.

One concern comes with providing additional support for the explanation of ROH. The manuscript clearly explains how ROH represents recent inbreeding. However, a citation or the theoretical underpinnings of these ideas would be beneficial when referencing specific numbers. Notably, on lines 200-202 "Long ROH (>15.2 Mb) occur due to close inbreeding (a common ancestor <3 generations ago). Short ROH (<5.7 Mb) occur due to shared ancestors further back in time (>8 generations ago)." This explanation provides a good explanation of how results can be interpreted but it is unclear where these numbers come from. As the manuscript correctly points out, and is a part of the novelty and importance of this work, these whole genome analyses in regards to conservation are still more novel. As the manuscript will likely serve as a guide for future work, accurately demonstrating the validity of the above claims is of particular importance. I think further supporting the details and interpretations of ROH is needed. Particularly as recent work (Runs of homozygosity: windows into population history and trait architecture, Ceballos et al. 2018) seems to indicate those moderate runs of ROH represent genetic drift. More on ROH below.

Additionally I have a few concerns that I don't think were thoroughly addressed, but I believe the authors could address them without any additional analyses. Specifically, in some instances the response in the cover letter was adequate but it was not carried into the manuscript itself. In the cover letter, the authors state that that they now refer to the divergence time between North and South America pumas to be a more general time around the "Late Pleistocene". However, they still put hard numbers in the abstract (200kya). I support the authors, as they stated in their cover letter, of using the 100–300kya time span for divergence and suggest replacing "~200,000" with "~100–300 kya" (and throughout).

Ln 80–98: Per the cover letter, the authors stated that they added a paragraph to the intro on the trade-off between the number of individuals sampled and the number of loci sequenced. However, the paragraph only properly addresses the benefits of whole genome analysis and doesn't discuss the cons of low sample sizes.

Ln 164–172: The authors acknowledge in their cover letter that the majority of their samples are from populations that are known to be isolated. However, they give nearly equal weight to the hypotheses that this could be the result of (1) isolation by distance and (2) true genetic structure. I would like

them to explicitly address this in the paper.

The sample size is extremely low for a thorough genetic structure analysis and the supplementary plots (Supp Fig 9) do not overwhelmingly support any specific K value. I am not convinced the sample size and distribution are adequate for a STRUCTURE analysis. If the authors insist on including STRUCTURE, it would be useful to display $K=2$ through $K=10$ so the reader can assess the hierarchical structure that almost certainly exists given the non-random sampling. I would also like to see mean and variance represented in Supp Fig 9 (top panel); not the maximum K. I would like to see those specific STRUCTURE plots reformatted.

Ln 177. As pointed out in the first round of reviews, ROHs arise from inbreeding AND genetic drift (Ceballos et al 2018. Nature Reviews Genetics); yet, genetic drift is nearly dismissed in this paper. The authors state: "Although genetic drift is a dominant evolutionary force in small populations, the strong correlation among linked sites that is characteristic of ROH requires close inbreeding, and would not be observed due to genetic drift alone." This statement either needs support via citations or genetic drift needs to be included in the discussion on the evolution of these isolated populations. Are the authors suggesting the ROHs were due to inbreeding alone? I don't think either evolutionary force caused the observed patterns alone. I suggest they properly incorporate drift into the manuscript along with inbreeding. For example, the literature indicates that pumas from SC and SMM area have a low effective population size and have experienced a relatively recent bottleneck, overwhelmingly indicating genetic drift (Gustafson et al 2018. Cons Gen). This is also the case in the Florida panther, which also had a recent founder effect from Texas pumas (Johnson et al 2010. Science).

Ln 228: SMM12 and SMM22 must correspond to P12 and P22 from the Riley et al. 2014 paper. If so, then it can be assumed that they are not directly related (based on the pedigree in that paper) and therefore should not necessarily share ROHs, as observed. However, SMM22 had a much larger proportion of its genome in ROH, suggesting drift and inbreeding. In contrast, SMM12 (perhaps a different population designation should be used than SMM) had values more similar to the YNP individual, suggesting P12 came from a larger population. This should be explicitly addressed. Also, what do the authors suspect the percent of shared ROHs would be for P12 and P12's offspring and then also with his inbred offspring.

Ln 299–310: The CA puma P12 needs to be explicitly addressed here. His genome resembles the YNP genome in many respects and he should not necessarily be lumped into the SMM group, especially since he is a known disperser. This paragraph would also be a good place to discuss genetic drift.

Ln 353–361: One concern is that the authors are undervaluing heterozygosity as a measure of population genetic health. The narrative comes off as negative towards the measure, which I don't think the authors are trying to be. Clearly it correlates well with the percent of genome in ROHs. Much like predicting outbreeding depression, the difference between heterozygosity and genome % in ROH for EVG21 could have been predicted. I would like to see the negativity toned down a little and perhaps a short discussion about: how the relationship may or may not be different when more than a single individual is measured from a population and how the relationship would look in the EVG after several generations.

Minor concerns

Ln 25: "...assisted gene flow followed by chromosomal crossing over would restore..."

Ln 26: "...a Florida panther that was descended from..."

Ln 166: replace "especially" with "also"

Ln 331–332. The road effect was also clearly demonstrated in the Santa Ana Range, CA (Gustafson et al 2017; Royal Society Open Science).

Reviewer #4 (Remarks to the Author):

The authors generated a genome assembly (PumCon1.0) of a wild male puma using Illumina and Oxford Nanopore reads. In overall, the authors nicely showed an example of how a newly built reference genome for a particular species can be served for population studies by including multiple individuals collected from different locations. Although the BUSCO scores and scaffold N50 shows good quality and contiguity, showing a few more details will help ensure the genome assembly is properly representing the species.

1. What is the expected genome size? This can be estimated from the illumina reads, let's say 31-mer or 45-mers that the authors used to build for their initial assembly. Running GenomeScope or preQC will give a better understanding of the expected haploid genome size, level of heterozygosity, and the repeat contents.
2. Add NG50 values below the N50 values using the estimated genome size obtained from above 1.
3. Hi-C interaction maps are good sources for validating structural accuracy of the genome. Although the authors have shown this in Supplementary Figure 1, the figure is hard to see when zoomed in. A larger figure with scaffold names labeled to each box will help better understand.
4. Include the illumina read depth coverage in the Results.
5. How does the read depth distribution look like? Include the illumina read mapping distribution for the PumCon1.0 assembly. As chrX was assembled with special care, and was used for downstream analysis, layering the chrX read distribution to the total read depth distribution will be useful to show the assembly went correct. Generating the distribution from 3C36 as well as SMM13.
6. Besides ROH, which uses SNPs, no structural concordance among the puma genomes are shown. Structural variant calling of the other 9 genomes against PumCon1.0 will be useful to show inbred / outbred differences. Structural variant calling from the same individual (3C36) will also detect any kind of misassembly.
7. Heterozygosity from the above 1. Can be also used as an indicator. Compare those to the SNP level heterozygosity measured in the Supplementary Table 3.

Reviewers' comments:

Reviewer #1 (Remarks to the Author):

In this manuscript, Saremi, Supple, et al. developed a chromosome-level genome assembly and resequenced the genomes of ten pumas across the Americas. This work provides the potential for important preliminary information on puma evolutionary history by conducting full genome sequencing on ten pumas across the range. Importantly, this marks the first published genome of *Puma concolor* which in itself is an important contribution to the scientific literature. This manuscript represents a monumental effort and has implications for the ecology, evolution, and conservation of pumas. A major strength of this manuscript is the vast quantity of sequence data. The construction of the genome was comprehensive, accounting for sex chromosomes, combining long reads with shorter high coverage Illumina runs.

In addition, the genomes of the Florida panther provide a unique opportunity to evaluate the effects of a genetic rescue event on the genome level. As noted previously, the genomes in themselves are a valuable addition to the literature. In addition, the authors have done a great job with the revision providing not only a clearer explanation of how results can be interpreted but highlighting alternative explanations. The response letter was well done and thoroughly addressed reviewer concerns.

We thank the reviewer for their enthusiasm for our paper, and for recognizing the amount of work involved!

One concern comes with providing additional support for the explanation of ROH. The manuscript clearly explains how ROH represents recent inbreeding. However, a citation or the theoretical underpinnings of these ideas would be beneficial when referencing specific numbers. Notably, on lines 200-202 “Long ROH (>15.2 Mb) occur due to close inbreeding (a common ancestor <3 generations ago). Short ROH (<5.7 Mb) occur due to shared ancestors further back in time (>8 generations ago).” This explanation provides a good explanation of how results can be interpreted but it is unclear where these numbers come from. As the manuscript correctly points out, and is a part of the novelty and importance of this work, these whole genome analyses in regards to conservation are still more novel. As the manuscript will likely serve as a guide for future work, accurately demonstrating the validity of the above claims is of particular importance. I think further supporting the details and interpretations of ROH is needed. Particularly as recent work (Runs of homozygosity: windows into population history and trait architecture, Ceballos et al. 2018) seems to indicate those moderate runs of ROH represent genetic drift. More on ROH below.

We thank the reviewer for pointing out this lack of clarity. The equation we used to convert ROH length to the number of generations was previously only included in the methods section. For clarity, we have also now included it in the results section, along with an additional reference that includes the theoretical basis of the equation. Our designation of

"short" and "long" is arbitrary, and used for ease of reading. The Ceballos et al. 2018 paper finds that "intermediate" ROH (defined as hundreds of kb to 2 Mb) may be explained by drift. We were unable to confidently call ROH <2 Mb without population estimates of allele frequencies. All our ROH analyses therefore equate to Ceballos's "long" ROH. We hope that our changes both clarify our approach and address these concerns

Additionally I have a few concerns that I don't think were thoroughly addressed, but I believe the authors could address them without any additional analyses. Specifically, in some instances the response in the cover letter was adequate but it was not carried into the manuscript itself. In the cover letter, the authors state that that they now refer to the divergence time between North and South America pumas to be a more general time around the "Late Pleistocene". However, they still put hard numbers in the abstract (200kya). I support the authors, as they stated in their cover letter, of using the 100–300kya time span for divergence and suggest replacing “~200,000” with “~100–300 kya” (and throughout).

We apologize for this oversight, and have made this change in the manuscript.

Ln 80–98: Per the cover letter, the authors stated that they added a paragraph to the intro on the trade-off between the number of individuals sampled and the number of loci sequenced. However, the paragraph only properly addresses the benefits of whole genome analysis and doesn't discuss the cons of low sample sizes.

We appreciate the reviewer's concerns that we have not adequately addressed this in the first revision. We have therefore added additional details on the trade-off between whole genomes and sample size, specifically that our use of whole genomes comes at the cost of spatial resolution. To this same end, we added a statement to the results that, because of our sampling scheme and choice of whole genomes (which allowed us to estimate ROH) we are unable to distinguish between the hypotheses of isolation by distance with structured sampling and true discrete genetic structure.

Ln 164–172: The authors acknowledge in their cover letter that the majority of their samples are from populations that are known to be isolated. However, they give nearly equal weight to the hypotheses that this could be the result of (1) isolation by distance and (2) true genetic structure. I would like them to explicitly address this in the paper.

The reviewer is correct that this was inadequately addressed in the revised version of our manuscript. We have now added a statement to the population structure results that, given our limited spatial sampling, we are unable to disentangle competing hypotheses. We feel, however, that distinguishing between these hypotheses is not the main goal of the manuscript, and hope that this point is conveyed more clearly and more accurately in the present version.

The sample size is extremely low for a thorough genetic structure analysis and the supplementary plots (Supp Fig 9) do not overwhelmingly support any specific K value. I am not convinced the sample size and distribution are adequate for a

STRUCTURE analysis. If the authors insist on including STRUCTURE, it would be useful to display K=2 through K=10 so the reader can assess the hierarchical structure that almost certainly exists given the non-random sampling. I would also like to see mean and variance represented in Supp Fig 9 (top panel); not the maximum K. I would like to see those specific STRUCTURE plots reformatted.

We agree with the reviewer that our geographic sampling limits our ability to determine the number of populations present in our sample. For this reason, we do not rely on the results of STRUCTURE to determine the correct K value, but instead use this analysis simply to highlight that genetic structure follows geography and, more critically to our manuscript, that EVG21 is of mixed ancestry.

We agree, however, that our K selection plots would benefit from reformatting, and now show the mean $L(K)$ and standard deviation in the top panel. The results of both methods of K selection (mean $L(K)$ and delta K) indicate that K=3 best explains our data. This result, while unsurprising given our sampling, follows those obtained by our other population analyses (PCA and TreeMix) and therefore provides additional support. We prefer therefore to retain the STRUCTURE panel in the main figure. However, because our sampling is not sufficiently powerful to estimate K, we do not indicate a best K value in our results, and have rephrased the methods section to indicate the limitations of proper cluster assignment given our sampling scheme. We have also included, as suggested by the reviewer, an additional supplementary figure (Supplementary Fig. 11) that shows results from STRUCTURE for K=2 through K=10. We hope that these changes address the reviewer's concerns.

Ln 177. As pointed out in the first round of reviews, ROHs arise from inbreeding AND genetic drift (Ceballos et al 2018. Nature Reviews Genetics); yet, genetic drift is nearly dismissed in this paper. The authors state: "Although genetic drift is a dominant evolutionary force in small populations, the strong correlation among linked sites that is characteristic of ROH requires close inbreeding, and would not be observed due to genetic drift alone." This statement either needs support via citations or genetic drift needs to be included in the discussion on the evolution of these isolated populations. Are the authors suggesting the ROHs were due to inbreeding alone? I don't think either evolutionary force caused the observed patterns alone. I suggest they properly incorporate drift into the manuscript along with inbreeding. For example, the literature indicates that pumas from SC and SMM area have a low effective population size and have experienced a relatively recent bottleneck, overwhelmingly indicating genetic drift (Gustafson et al 2018. Cons Gen). This is also the case in the Florida panther, which also had a recent founder effect from Texas pumas (Johnson et al 2010. Science).

We thank the reviewer for highlighting this once again. We elected not to include a full exploration of the effect of drift on ROH because we are unable to identify and therefore compare ROH that are likely caused by drift that is distinguishable from inbreeding. To clarify this, we now state in the results that our ROH analyses focus on ROH >2 Mb. Consistent with our statement that these ROH would not be observed due to drift alone, Ceballos et al. 2018 states "long ROH (over 1–2 Mb) [arise] from recent parental relatedness". Ceballos et al refers to drift affecting ROH in the "hundreds of kb to 2 Mb" range, which, as we stated previously is a range that we were unable to examine confidently (lacking a full population

panel). We have added the Ceballos citation, as well as Pemberton 2012, which states that "long ROH measuring multiple Mb... probably result from recent parental relatedness".

Ln 228: SMM12 and SMM22 must correspond to P12 and P22 from the Riley et al. 2014 paper. If so, then it can be assumed that they are not directly related (based on the pedigree in that paper) and therefore should not necessarily share ROHs, as observed. However, SMM22 had a much larger proportion of its genome in ROH, suggesting drift and inbreeding. In contrast, SMM12 (perhaps a different population designation should be used than SMM) had values more similar to the YNP individual, suggesting P12 came from a larger population. This should be explicitly addressed. Also, what do the authors suspect the percent of shared ROHs would be for P12 and P12's offspring and then also with his inbred offspring.

The reviewer is correct that SMM12 and SMM22 correspond to P12 and P22 from Riley et al. 2014. This renaming is indicated in Supplementary Table 3. We have added the specific references (e.g. Riley et al. 2014) to this table for clarity.

These two individuals currently reside in the Santa Monica Mountains south of the 101 freeway. P12 is known to have originated north of the 101 freeway, which subdivides this population. We have chosen to give P12 the SMM designation because it is where this animal currently resides, and is therefore the population to which he has the potential to contribute genetically. In addition, as P12 and P22 originated only a few miles apart from each other and given our coarse geographic sampling, we have chosen to treat these two individuals as coming from a single population. We note that paragraph nine of the Discussion section goes into more detail regarding the subdivision within this population and the genomic implications of this separation.

Ln 299–310: The CA puma P12 needs to be explicitly addressed here. His genome resembles the YNP genome in many respects and he should not necessarily be lumped into the SMM group, especially since he is a known disperser. This paragraph would also be a good place to discuss genetic drift.

While the genome of SMM12 does resemble YNP198 in many respects, SMM12 does have more of the larger ROH that are clearly attributable to inbreeding (Figure 4). SMM12's origins are discussed explicitly in the Discussion, and in the response above. We have added a new statement to paragraph nine of the discussion regarding the differences in ROH distributions between SMM12 and SMM22.

As discussed above, the large size of the ROH (>2 Mb) we examined do not allow us to examine the effects of drift that is separate from inbreeding.

Ln 353–361: One concern is that the authors are undervaluing heterozygosity as a measure of population genetic health. The narrative comes off as negative towards the measure, which I don't think the authors are trying to be. Clearly it correlates well with the percent of genome in ROHs. Much like predicting outbreeding depression, the difference between heterozygosity and genome % in ROH for EVG21 could have been predicted. I would like to see the negativity toned down a little and perhaps a short discussion about: how the relationship may or may not be different when more

than a single individual is measured from a population and how the relationship would look in the EVG after several generations.

We appreciate the reviewer's comment on this point. We agree that the difference between heterozygosity and the percent of the genome in ROH could be predicted given the known admixture in the history of the Everglades population. Our argument, which we did not intend to be overly negative, is that caution is warranted when using average genome-wide heterozygosity because population history is rarely as well documented as it was in the Florida panthers. Our broad geographic sampling allowed us to detect the historical admixture without ROH analyses, but sampling will not always be sufficient to detect admixture from genetic-scale data. Given a population of conservation concern with an unknown history, researchers might be unaware that recent admixture has affected the estimates of average genome-wide heterozygosity. To this end, we state that "While in some species average genome-wide heterozygosity is highly correlated with the level of inbreeding estimated using ROH, in systems with admixture, average heterozygosity estimates can be deceptive." We hope that this statement provides a sufficiently balanced view while at the same time highlighting the novelty of our results relating to the value of whole genome data sets.

Minor concerns

Ln 25: "...assisted gene flow followed by chromosomal crossing over would restore..."

We appreciate the reviewer's recommendation, but believe that 'chromosomal crossing over' is implied in gene flow and simpler to understand. We will keep the sentence as:

"... homozygosity were rarely shared among puma populations, suggesting that assisted gene flow would restore local genetic diversity."

Ln 26: "...a Florida panther that was descended from..."

We thank the reviewer for their input, however, to lessen wordiness, we have chosen to keep the text as below. We interpret the meaning of both sentences identically.

"The genome of a Florida panther descended from ..."

Ln 166: replace "especially" with "also"

We respectfully disagree with this suggestion. We are attempting to highlight an important caveat in our population structure results by emphasizing a likely alternative scenario. We have chosen to keep the text as below.

"This artefact is especially likely to occur in geographically widespread samples from well connected species that show genetic isolation by distance, where limited dispersal results in the accumulation of local genetic variants"

Ln 331–332. The road effect was also clearly demonstrated in the Santa Ana Range, CA (Gustafson et al 2017; Royal Society Open Science).

We thank the reviewer for their citation recommendation and have added the citation.

Reviewer #4 (Remarks to the Author):

The authors generated a genome assembly (PumCon1.0) of a wild male puma using Illumina and Oxford Nanopore reads. In overall, the authors nicely showed an example of how a newly built reference genome for a particular species can be served for population studies by including multiple individuals collected from different locations. Although the BUSCO scores and scaffold N50 shows good quality and contiguity, showing a few more details will help ensure the genome assembly is properly representing the species.

1. What is the expected genome size? This can be estimated from the illumina reads, let's say 31-mer or 45-mers that the authors used to build for their initial assembly. Running GenomeScope or preQC will give a better understanding of the expected haploid genome size, level of heterozygosity, and the repeat contents.

We thank the reviewer for this suggestion to provide greater clarity of the quality of our genome assembly. As recommended, we ran Jellyfish using a kmer of 45 and estimated the genome size using GenomeScope.

2. Add NG50 values below the N50 values using the estimated genome size obtained from above 1.

As suggested by the reviewer, we have added a row in Supplementary Table 1 which indicates the NG50 scaffold size for each assembly version given this expected genome size.

3. Hi-C interaction maps are good sources for validating structural accuracy of the genome. Although the authors have shown this in Supplementary Figure 1, the figure is hard to see when zoomed in. A larger figure with scaffold names labeled to each box will help better understand.

We agree with the reviewer that we could improve upon the resolution of the previous Hi-C contact map. We have updated Supplementary Figure 1 by creating a higher resolution contact map for the Hi-C data using the Hi-C scaffolded assembly. We have also noted the scaffolds on the image as suggested by the reviewer.

4. Include the illumina read depth coverage in the Results.

We thank the reviewer for this suggestion. We previously only noted the shotgun coverage in the Methods section. We have updated the text to include the read depth coverage of the sequencing data in the Results section.

5. How does the read depth distribution look like? Include the illumina read mapping distribution for the PumCon1.0 assembly. As chrX was assembled with special care, and was used for downstream analysis, layering the chrX read distribution to the total read depth distribution will be useful to show the assembly went correct. Generating the distribution from SC36 as well as SMM13.

We thank the reviewer for this recommendation to add plots that validate our X chromosome assembly. We have included coverage density plots for the three X chromosome scaffolds, along with one autosomal scaffold for SC36 and SMM13 as Supplementary Figure 2.

6. Besides ROH, which uses SNPs, no structural concordance among the puma genomes are shown. Structural variant calling of the other 9 genomes against PumCon1.0 will be useful to show inbred / outbred differences. Structural variant calling from the same individual (SC36) will also detect any kind of misassembly.

We agree with the reviewer that examining for structural variants in the reference genome is important for overall genome quality assessment, especially given their possible effect on downstream analyses. However, visual inspection of structural variants is also arduous, and would require a large investment of time for this sole purpose. Given that we made only one Hi-C library from one puma, and that we did not use our long-read data for scaffolding purposes, we can presume that our puma genome does contain some incorrect scaffold orientations, as do most assemblies. However, we believe, based on the high mapping percentages of all the other pumas in our panel, that the number of structural variants is low, and thus would not affect our population level analyses results.

7. Heterozygosity from the above 1. Can be also used as an indicator. Compare those to the SNP level heterozygosity measured in the Supplementary Table 3.

As discussed above, structural variant calling is beyond the scope of the current paper. We filtered our SNPs to remove suspicious variants, including those caused by structural variants or misassemblies. We also used multiple methods to call heterozygosity based on our filtered SNPs. The different methods were consistent. Additionally, we used multiple methods to call ROH and confirmed visually that the method we selected performed well. Thus we believe that any possible structural variants in the assembly will not result in a large fluctuation from our current heterozygosity estimates.

Reviewers' comments:

Reviewer #4 (Remarks to the Author):

The authors improved the supplementary figures and tables to support better understanding regarding the assembly quality. Although few concerns have been well addressed, there are newly raised concerns about the assembly quality and the claim for being inbred and less heterozygous.

1. The GenomeScope results are missing. What is the estimated level of heterozygosity? The k-mer histogram are a very good indicator for measuring the level of heterozygosity. Inbreeding and outbreeding is easily detectable, independent from SNP or SV based measurements. Concluding the heterozygosity level solely on SNPs is very limited, especially on assemblies generated from less contiguous contigs. k=21 or k=31 will be suitable for this analysis. The illumina read coverage from Supplementary Table 3 will be suitable for GenomeScope except BR406. BR506 might fail to converge due to low depth of coverage. If GenomeScope fails to converge, it is still fine, including the k-mer histogram will show the level of heterozygosity. The more outbred, there will be 2 peaks shown - one for the heterozygous region, half the expected coverage and one for the homozygous region. The more inbred, the more likely to see 1 peak near the expected coverage.

2. Claiming this genome as being 'high quality' seems rather ambitious. What is the final contig N50 post gap filling and polishing? This should be included in the Results and Supplementary Table 1. The authors find SV analysis beyond the scope and admit their assembly contain some misassembly. To claim being 'high quality', the authors should include a curation step to detect misassemblies, fix those, and validate with an independent platform. As the initial contig size was only 36.6kb, it is difficult to imagine the assembly being reliable for complex repetitive region. If this is out of scope, the authors should revise the term 'high quality' to 'draft'.

3. How are the QV assessed? The reference citing is only referring to the tool names, with no details. Depending on how the variants (errors) are called, the QV may vary. The authors should include how this was performed with exact QV numbers. Was it exactly 40?

4. The scaffolds are large, and some of them are reaching chromosome-scale according to the Hi-C plot and synteny to the domestic cat genome as in Supplementary Figure 1 and 3. However, there are notable number of scaffolds that are left and could be joined based on the Hi-C maps. To use the term 'chromosome-scale', the scaffolds should be joined and chromosome assigned. For example, scaffold_2176 and scaffold_942 can be joined by inverting scaffold_942 and placing it in front of scaffold_2176. It seems feasible to join the scaffolds >20M based on the Hi-C evidence. Having a chromosome assigned draft assembly will be extremely helpful to the research community doing synteny based comparative analysis.

Reviewer #4 (Remarks to the Author):

The authors improved the supplementary figures and tables to support better understanding regarding the assembly quality. Although few concerns have been well addressed, there are newly raised concerns about the assembly quality and the claim for being inbred and less heterozygous.

1. The GenomeScope results are missing. What is the estimated level of heterozygosity? The k-mer histogram are a very good indicator for measuring the level of heterozygosity. Inbreeding and outbreeding is easily detectable, independent from SNP or SV based measurements. Concluding the heterozygosity level solely on SNPs is very limited, especially on assemblies generated from less contiguous contigs. k=21 or k=31 will be suitable for this analysis. The illumina read coverage from Supplementary Table 3 will be suitable for GenomeScope except BR406. BR506 might fail to converge due to low depth of coverage. If GenomeScope fails to converge, it is still fine, including the k-mer histogram will show the level of heterozygosity. The more outbred, there will be 2 peaks shown - one for the heterozygous region, half the expected coverage and one for the homozygous region. The more inbred, the more likely to see 1 peak near the expected coverage.

The reviewer brings up a good point regarding quantifying genome heterozygosity using a non-reference biased technique. We previously calculated heterozygosity using two different variant based methods, both of which require the use of a reference to obtain estimates (Supplementary Table 3).

As suggested by the reviewer, we ran GenomeScope to estimate heterozygosity for all ten pumas using a k=21, as per the recommendation of the GenomeScope authors (Vurture et al., 2017). We include the results of this analysis for each puma in the figure below. Most of the ten pumas show only one significant peak in their kmer frequency distribution, likely the result of both low levels of heterozygosity and moderate genome coverage. Only at coverages >45x do we begin to discern the two peaks. Our highest coverage sample, CYP51 with 55x coverage, shows two discernible peaks in the GenomeScope plot. Based on ancestry records, we have estimates of the inbreeding level of CYP51, and thus we believe it is one of the least heterozygous pumas in our samples. We believe the heterozygosity estimates for the other pumas are significantly underestimated using this approach, due to the inability to identify the heterozygous peak.

As these results are inconclusive, we elected not to add them to the main text, but hope that the reviewer and editors find their inclusion useful here.

2. Claiming this genome as being ‘high quality’ seems rather ambitious. What is the final contig N50 post gap filling and polishing? This should be included in the Results and Supplementary Table 1. The authors find SV analysis beyond the scope and admit their assembly contain some misassembly. To claim being ‘high quality’, the authors should include a curation step to detect misassemblies, fix those, and validate with an independent platform. As the initial contig size was only 36.6kb, it is difficult to imagine the assembly being reliable for complex repetitive region. If this is out of scope, the authors should revise the term ‘high quality’ to ‘draft’.

We have included the contig N50 for each version of the puma assembly in Supplementary Table 1, as suggested by the reviewer. We agree with the reviewer that the phrase “high quality” is subjective and unnecessary. We have gone through the manuscript and removed this phrase, replacing it where necessary with “draft,” as is standard.

3. How are the QV assessed? The reference citing is only referring to the tool names, with no details. Depending on how the variants (errors) are called, the QV may vary. The authors should include how this was performed with exact QV numbers. Was it exactly 40?

We thank the reviewer for catching this, and we have removed reference to QV from the manuscript, as it should not have been included (this was a misunderstanding on our end). As our shotgun genome is a de Bruijn graph assembly, the most accurate way to assess QV is to compare the assembly to independent dataset, such as a BAC library. Unfortunately, no such libraries exist for the puma. Prior research has estimated QV for a genome assembly with haplotype based methods, using only the shotgun reads which are also used to generate the assembly (Bickhart et al., 2017, Gao et al., 2018). However since this estimation is not truly independent from the genome assembly, we do not feel this is an appropriate method to use for our genome.

4. The scaffolds are large, and some of them are reaching chromosome-scale according to the Hi-C plot and synteny to the domestic cat genome as in Supplementary Figure 1 and 3. However, there are notable number of scaffolds that are left and could be joined based on the Hi-C maps. To use the term ‘chromosome-scale’, the scaffolds should be joined and chromosome assigned. For example, scaffold_2176 and scaffold_942 can be joined by inverting scaffold_942 and placing it in front of scaffold_2176. It seems feasible to join the scaffolds >20M based on the Hi-C evidence. Having a chromosome assigned draft assembly will be extremely helpful to the research community doing synteny based comparative analysis.

As with our response to comment #2 above, the phrase “chromosome-scale” is subjective and not particularly useful, especially given that all of the statistics referencing the quality of the genome are provided. We have removed the phrase from the text. We now refer to our genome as a draft genome and leave further polishing and chromosome assignment for future work.

REVIEWERS' COMMENTS:

Reviewer #4 (Remarks to the Author):

Dear authors,

I find the k-mer heterozygosity very well matching the observed SNP-level heterozygosity. It is well representing the reference-free heterozygosity estimate, matching the heterozygosity in Supplementary Table 3. I see it as a compelling evidence for the SNP-level heterozygosity estimate, which is the base line of all analysis performed in this manuscript. One possible suggestion would be to put a note for EVG21 and SC29. Usually, the k-mer level is lower in general for females compared to males. The greater heterozygosity observed in males comes from X and Y chromosome difference. Thus, comparison should be made between identical genders. When comparing between male samples, the trend matches from what has been observed. In the same line, EVG21 is more homozygous compared to SC29.

SNP-based heterozygosity measures are biased towards the reference. If some part of the target genome is too divergent from the reference, it can't be mapped, thus will be excluded. In the same context, if the reference is incomplete and is missing sequences, in illumina-only assemblies, those will be never examined. Therefore, adding a reference-free analysis result showing the heterozygosity pattern matching the SNP-based result will certainly improve the reliability of this study. I would strongly suggest adding a column for k-mer level heterozygosity in Supplementary Table 3.

One final request is to take out the sentence in Discussion, line 246-247:

"This approach presents a less costly alternative to recovering missing sequence than alternate strategies that begin, for example, with high-coverage long-read data."

No direct comparison has been made to an equivalent species' high-coverage long-read data assembly. 140Mbp of gap filled sequences does not always guarantee to be "correct". Not enough evidence to draw such conclusion and is out of scope.

I find the rest of my concerns are well addressed.

Dear Lin,

Thank you for your continued consideration of our manuscript. Below, please find our response to reviewer four's additional comments.

Best regards,

Beth, Nedda, and Megan

REVIEWERS' COMMENTS:

Reviewer #4 (Remarks to the Author):

Dear authors,

I find the k-mer heterozygosity very well matching the observed SNP-level heterozygosity. It is well representing the reference-free heterozygosity estimate, matching the heterozygosity in Supplementary Table 3. I see it as a compelling evidence for the SNP-level heterozygosity estimate, which is the base line of all analysis performed in this manuscript. One possible suggestion would be to put a note for EVG21 and SC29. Usually, the k-mer level is lower in general for females compared to males. The greater heterozygosity observed in males comes from X and Y chromosome difference. Thus, comparison should be made between identical genders. When comparing between male samples, the trend matches from what has been observed. In the same line, EVG21 is more homozygous compared to SC29.

SNP-based heterozygosity measures are biased towards the reference. If some part of the target genome is too divergent from the reference, it can't be mapped, thus will be excluded. In the same context, if the reference is incomplete and is missing sequences, in illumina-only assemblies, those will be never examined. Therefore, adding a reference-free analysis result showing the heterozygosity pattern matching the SNP-based result will certainly improve the reliability of this study. I would strongly suggest adding a column for k-mer level heterozygosity in Supplementary Table 3.

We created a new Supplementary Table 4 that includes the heterozygosity estimates from GenomeScope, as suggested. We included a note about the role of genome coverage and heterozygosity in discerning the two peaks in the kmer plot from GenomeScope and about the potential influence of biological sex on the GenomeScope heterozygosity values.

One final request is to take out the sentence in Discussion, line 246-247:

“This approach presents a less costly alternative to recovering missing sequence than alternate strategies that begin, for example, with high-coverage long-read data.”

No direct comparison has been made to an equivalent species' high-coverage long-read data assembly. 140Mbp of gap filled sequences does not always guarantee to be “correct”. Not enough evidence to draw such conclusion and is out of scope.

We have removed this sentence from the main text as suggested.

I find the rest of my concerns are well addressed.

We thank the reviewer for their continued careful reading of our manuscript.